# Synthesis of Dense MgB_2_ Superconductor via In Situ and Ex Situ Spark Plasma Sintering Method

**DOI:** 10.3390/ma14237395

**Published:** 2021-12-02

**Authors:** Joseph Longji Dadiel, Sugali Pavan Kumar Naik, Paweł Pęczkowski, Jun Sugiyama, Hiraku Ogino, Naomichi Sakai, Yokoyama Kazuya, Tymon Warski, Anna Wojcik, Tetsuo Oka, Masato Murakami

**Affiliations:** 1Superconducting Materials Laboratory, Graduate School of Science and Engineering, Shibaura Institute of Technology, 3-7-5 Toyosu, Koto-Ku, Tokyo 135-8548, Japan; mb18017@shibaura-it.ac.jp (J.S.); h-ogino@aist.go.jp (H.O.); okat@sic.shibaura-it.ac.jp (T.O.); masatomu@shibaura-it.ac.jp (M.M.); 2Department of Physics, Tokyo University of Science, 1 Chome-3 Kagurazaka, Shinjuku City, Tokyo 162-8601, Japan; spavankumarnaik@yahoo.in; 3Research Institute for Advanced Electronics and Photonics, National Institute of Advanced Industrial Science and Technology (AIST), 1-1-1 Central 2, Umezono, Tsukuba 305-8568, Japan; 4Institute of Physical Sciences, Faculty of Mathematics and Natural Sciences, School of Exact Sciences, Cardinal Stefan Wyszyński University, K. Wóycickiego 1/3 Street, 01-938 Warsaw, Poland; p.peczkowski@wp.pl or; 5Department of Electrical and Electronics, Faculty of Engineering, Ashikaga University, 286-1 Omae-cho, Ashikaga-Shi, Tochigi 326-8558, Japan; k-yokoyama@ashitech.ac.jp; 6Łukasiewicz Research Network, Institute of Non-Ferrous Metals, Sowinskiego 5 Street, 44-100 Gliwice, Poland; tymon.warski@imn.gliwice.pl; 7Department of Engineering Materials and Biomaterials, Faculty of Mechanical Engineering, Silesian University of Technology, Konarskiego 2a Street, 44-100 Gliwice, Poland; 8Institute of Metallurgy and Materials Science, Polish Academy of Science, Reymonta 25 Street, 30-059 Krakow, Poland; a.wojcik@imim.pl

**Keywords:** MgB_2_, spark plasma sintering, microstructure, flux pinning, critical current density, grain connectivity

## Abstract

In this study, high-density magnesium diboride (MgB_2_) bulk superconductors were synthesized by spark plasma sintering (SPS) under pressure to improve the field dependence of the critical current density (*J*_c_-*B*) in MgB_2_ bulk superconductors. We investigated the relationship between sintering conditions (temperature and time) and *J*_c_-*B* using two methods, ex situ (sintering MgB_2_ synthesized powder) and in situ (reaction sintering of Mg and B powder), respectively. As a result, we found that higher density with suppressed particle growth and suppression of the formation of coarse particles of MgB_4_ and MgO were found to be effective in improving the *J*_c_-*B* characteristics. In the ex situ method, the degradation of MgB_2_ due to pyrolysis was more severe at temperatures higher than 850 °C. The sample that underwent SPS treatment for a short time at 850 °C showed higher density and less impurity phase in the bulk, which improved the *J*_c_-*B* properties. In addition, the in situ method showed very minimal impurity with a corresponding improvement in density and *J*_c_-*B* characteristics for the sample optimized at 750 °C. Microstructural characterization and flux pinning (*f*_P_) analysis revealed the possibility of refined MgO inclusions and MgB_4_ phase as new pinning centers, which greatly contributed to the *J*_c_-*B* properties. The contributions of the sintering conditions on *f*_P_ for both synthesis methods were analyzed.

## 1. Introduction

Since 1953, intermetallic MgB_2_ with a hexagonal structure has been known [1]. Its superconducting critical temperature (*T*_c_), up to 39 K, was observed in 2001 due to the presence of two distinct superconducting gaps [2]. The inexpensive cost, strong mechanical properties [3,4], and long coherence length of this material make it ideal for a variety of applications [5]. It has a greater upper critical field than standard NbTi and Nb_3_Sn superconductors. It has fewer anisotropic effects and no weak links as compared to superconducting cuprates. The grain boundaries do not act as weak links that inhibit superconducting current; therefore, this makes it possible to fabricate superconducting bulk and wires in crystalline forms [5,6].

The MgB_2_ compound has been found to be one of the reliably promising materials for the next generation of superconductors due to its cost and relative density of 2.63 g/cm^3^, as well as its ease of manufacture due to its suitability for light-weight applications [1]. In the literature [6,7], we can find reports of research on this topic: the effects of the SPS temperature on the mechanical properties of the MgB_2_ bulk superconductor were investigated through bending tests of specimens cut from the bulk samples. Both the Young modulus and bending strength were improved by an increase in the SPS temperature [8]. For better practical application, together with the superconducting properties of MgB_2_, the critical current density (*J*_c_) is important as it improves their mechanical properties [9]. However, since MgB_2_ is polycrystalline in nature, most techniques for producing this material involve low relative densities that have poor material performance [10]. There is a solid relationship between the super-current conduction region and the material density, the resulting effect of which is constantly reflected in *J*_c_ [11,12]. Three different advances are used for the production of MgB_2_ wires and bulk samples, hence: the ex situ technique [9,13,14], in situ [9,14,15,16,17], and the Mg Internal Diffusion (IMD) technique [18]. Gajda et al. [8,9] studied the fabrication of MgB_2_ wire by hot isostatic pressure (HIP) and the influence of various physicochemical parameters on their superconducting properties. By applying SPS to fabricate sintered polycrystalline MgB_2_ bulks in our research, the bulk superconducting performance has been studied in relation to our starting powders and techniques. The approach to target the *J*_c_ of the samples, by enhancing it as a result of the controlled microstructure and improved density and flux pinning, was achieved by optimizing the sintering temperature conditions for both in situ and ex situ SPS processing with reduced deterioration of the MgB_2_ phase. The SPS process can be used to obtain full density without grain growth at a high heating rate (up to 600 °C/min or more) and a short holding time, usually a few minutes. The SPS process obtains fully dense samples at relatively low sintering temperatures, usually several hundred degrees lower than in the case of normal hot pressing [19]. However, there are difficulties with this method. Several authors have reported about the problem of MgB_2_ decomposition during the densification at high temperatures while sintering via ex situ methods [20,21,22]. The MgB_2_ bulk samples produced via the SPS method contain nano-sized Mg–B–O and higher borides (of compositions near MgB_4_, MgB_7_, MgB_12_, MgB_17_, and MgB_20_), as inclusions that can be pinning centers. The quantity of secondary phase inclusions is intensified at temperatures above 900 °C, especially when ground powder (ex situ) is used. Guo et al. [23] conducted thermo-analytical studies, in which they found the onset of oxidation at 600 °C and decomposition to higher borides at 930 °C. An additional positive aspect of the SPS is its flexibility in controlling current and temperature, which is an advantage in the successive heating and cooling rates. In the first place, it was expected that this would provide good control of grain growth and, to some extent, evaporation of Mg. For MgB_2_, a higher density of grain boundaries associated with a nano-structured material is desired for improved interfacial defect density and, hence, the superior flux pinning [24].

The research conducted so far has focused on the effect of the addition of nanoscopic diamond particles [25], silver [26], and rare earth particles [27] for improving the *J*_c_ of polycrystalline MgB_2_. The optimal amount of nano-scale addition of diamond and Ag to MgB_2_ enables the formation of high-density nano-inclusions in the MgB_2_ matrix and dramatically improves *J*_c_. In the current manuscript, we investigated the properties of high-density MgB_2_ superconductors prepared using the in situ and ex situ SPS techniques. The phase and microstructure of the samples were determined by X-ray diffraction (XRD) and field-emission scanning electron microscopy (FE-SEM). Recent reports have shown the role of sintering kinetics and precursor powders on improving flux pinning and *J*_c,S_ [28]. Besides the advantage of the *T*_c_, (which is around 39 K, as discussed earlier), the most important property to improve is the *J*_c_ and the upper critical fields *H*_c2_. In order to achieve this, it is important to improve the pinning force and the connectivity between particles by improving the density of the MgB_2_ bulk material. The explanation for our results also resides in the reduction of the cost of the application and processing time, we tried to synthesize the bulk MgB_2_ samples via both the ex situ and in situ SPS methods, by using MgB_2_ powder and mixing the stoichiometry ratios of Mg and B, respectively. This also aimed at fabrication of the bulk with abundant amounts of the grain-connectivity, along with high mass density, due to the respective reactions that could support even larger *J*_c,S_ in the MgB_2_ bulk samples, in both cases. The purpose of our study is to clarify the necessary conditions for high *J*_c_ using the SPS process. Here, we investigated the effects of the fabrication conditions on the crystal phase, microstructure, *J*_c_, and flux pinning of the high density bulk in both the SPS in situ and ex situ processes, clarified the differences in the SPS processes, and investigated the method for optimizing each fabrication condition to achieve high *J*_c_.

In this manuscript, we have focused on the production of MgB_2_ by spark plasma synthesis (SPS) and physico-chemical, magnetic, and superconducting properties analysis. We emphasize that the SPS is a promising way to produce dense samples with improved grain bonding and better MgB_2_ superconductor bulk densities, which could be an important variant in magnetic applications; therefore, we have dealt with this method in our work. The optimization of several parameters, such as dwell time, applied pressure, controlled heat, and current application rates, can effectively improve the immobilization of the stream and, hence, the efficiency of *J*_c_, as discussed in this manuscript. A comparative study would enable us understand more about the effectiveness of our approach on both the ex situ and in situ processing techniques.

## 2. Experimental Section

Three sets of samples were synthesized and considered for this study. **The first** series of samples were fabricated by SPS with a commercially available (MgB_2_, purity >97%, 100 meshes) powder obtained commercially from Kojundo Chemicals (Laboratory Co., Ltd., Saitama, Japan). In the ex situ process, the admixed MgB_2_ powder was inserted into the graphite die under ambient atmosphere and then charged into the SPS sample chamber similar to in situ synthesis. The graphite die was heated within the ranges of 800–1000 °C with the same rate of 50 °C min^−1^ by simultaneous application of current pulses for 15 min of dwell time. Further, the dwell time was optimized by varying for 1–10 min for the sample fabricated at optimal sintering temperature of 850 °C. At the end of each process, all samples were then furnace-cooled to room temperature (RT). Final SPS ex situ products were referred as ex situ-*W-Z* where *W* and *Z* represent the sintering temperatures and the dwell times, respectively). In the overall process, after removing the bulk sample from the SPS chamber, the graphite layer and the carbon-contaminated surfaces were removed, via polishing by SiC papers. The relative bulk densities of all the samples were estimated using the theoretical density of MgB_2_. **The second** series of samples investigated in this work were prepared by the in situ SPS technique. For the in situ SPS process, 0.6 g of the Mg (99.9%, 325 mesh) and B (98.5%, 250 nm) powder obtained from Kojundo Chemicals were mixed in the ratio of 1:2 and were loaded into a *φ*10 mm graphite die in a glovebox filled with high pure Ar. The schematic of the SPS sample chamber is shown in Figure 1. The graphite die has an inner diameter of 10 mm. It was lined with boron nitrite (BN) and was placed in the SPS apparatus (SS Alloy Co., Ltd., Higashi Hiroshima, Japan). The schematic diagram in Figure 1 also shows the cross section of the graphite die and its components. The sample chamber was evacuated and, then, filled with argon gas to a pressure of 0.5 atm. A pressure of 50 MPa was applied to the filled powder through the graphite die. The heating on the graphite commenced with the sintering temperatures being recorded for 720–775 °C ranges at the rate of 50 °C min^−1^, and these sets of samples were named as in situ-*X-Y* where *X* and *Y* are the sintering temperatures and dwell times, respectively. **The third** series of samples considered in this work was adopted from our previous work for the purpose of comparison with the SPS technique. The details for the experimental procedure for the in situ solid-state reaction and results are available in [27,28].

To identify any impurity phases formed during sintering process, the crystal structures and constituent phases of the fabricated MgB_2_ samples were investigated using a high-resolution X-ray powder diffractometer (SmartLab, Rigaku, Tokyo, Japan) using Cu-K_α_ radiation (λ = 1.5418 Å). The XRD patterns were collected with a step size of 0.02° over a 2*θ* range from 20° to 80°. Small sections were cut from the bulk pellets using a diamond slow saw and polished using grinding paper sizes, ranging from 400 to 1200 grit. The sample breakthroughs were also investigated using a FE-SEM (Jeol JSM-7100F model, Nippon Electronics, Tokyo, Japan). Further microstructural and elemental characterizations were carried out by transmission electron microscope (TEM Titan Themis G2 80-200 kV X-FEG, Thermo Fisher Scientific, Waltham, MA, USA). Field and temperature dependent characterizations were carried out. Magnetization hysteresis (*M*-*H*) loops were measured in field range from 0 to 5 T at 20 K, and temperature dependance magnetization curves were recorded using a superconducting quantum interference device (SQUID) magnetometer (MPMS V model, Quantum Design, San Diego, CA, USA). Temperature dependence measurements were carried out in the zero field cooled (ZFC) regime, whereby the system is cooled with no applied field to lowest temperature. The temperature was then swept from 10 K to 50 K after applying a magnetic field of 1 mT. Sub-specimens of typical dimensions of ~2.0 × 2.0 × 1.0 mm^3^ were collected from each sample and were utilized for *M*-*H* and *M*-*T* measurements. The *J_c_* values were estimated from the *M*-*H* loop based on the extended Bean critical state model using the relation:
(1)$$J_{C}\left(H\right)=\frac{20\times \Delta M\left(H\right)}{a^{2}\times d\left(b-\frac{a}{3}\right)}$$

The parameters in the relation can be summarized where *d* denotes the sample thickness, *a* and *b* are the cross sectional dimensions with *b* ≥ *a*, and *ΔM* is the difference of the magnetic moments during increasing and decreasing field in *M*-*H* loop [29].

## 3. Results and Discussion

### 3.1. Ex Situ Synthesis

The XRD patterns for the MgB_2_ bulks produced by ex situ sintering at various sintering temperatures from 800–1000 °C for 15 min are shown in Figure 2. The XRD also show a small amount of MgO in the starting powder, which was included for comparison. The X-ray diffraction patterns of all the samples show that the highest intensity Bragg peaks are representing the MgB_2_ phase as major. However, as the sintering temperature increased, the decomposition increased mainly to MgB_4_ and MgO as impurity phases. The presence of MgO and MgB_4_ impurity phases is not uncommon in MgB_2_ processing, and the relatively high contents of the observed impurity phases in these samples are related to the high processing temperatures used [23,24]. A sequel to the updated phase diagram of Mg–B system at high temperature, the Mg vapor is present in equilibrium with MgB_2_ or MgB_4_ [30]. The observed increase in the contents of MgB_4_ and MgO is expected if MgO forms from reaction of the Mg vapor [20]. However, there is the tendency of resolving this issue by improving the quality of the starting powder; hence, by carrying out powder processing in an inert dry atmosphere or by possible reduction in the processing temperature [20,21]. The decomposition was observed to intensify when the sintering temperature was increased above 850 °C. However, compared to the reported literature [31], the effect of the low proportions of the formed secondary phase was seen in the improvement of the flux pinning characteristics due to the refinement of the grains, which was articulated later in the sample microstructure.

The polycrystalline MgB_2_ sample density was determined to be 1.73 g/cm^3^, and the corresponding relative densification value is 66%. The density of the ex situ - 800 °C - 15 min to ex situ - 1000 °C - 15 min samples was found to be 2.41 g/cm^3^–2.59 g/cm^3^, which corresponds to 92–99% of the theoretical density of the same compound. The densification values are given in Table 1. It was observed that as the sintering temperature increased, the relative density values were increased gradually, which indicates the decrease of grain boundary in the bulk samples. The density of the SPS bulk samples is improved about 30%, which implies that grain connectivity is improved tremendously in the case of SPS MgB_2_ bulk samples. The achieved density in the MgB_2_ bulk samples is comparable with those reported by the literature on the same compound fabricated by SPS and even hot-press [32]. With the present results, we successfully fabricated highly dense MgB_2_ bulk samples using the SPS technique.

To further control the decomposition that occurred at 850 °C in the SPS MgB_2_ bulk sample, the dwell time was varied and the observed XRD patterns are shown in Figure 3. It was observed that as the dwell time decreased from 15 min to 1 min, the decomposition was effectively decreased. The densification values are shown in Table 1. These are the calculated values of the relative bulk densities. As the dwell time is controlled for each stage of synthesis from 15 min to 1 min, the densification was decreased from 97% to 86%, indicating that the dwell time is also an important parameter. With the resent results, it can be observed that the sintering temperature of 850 °C and a 10 min dwell is the best condition, which shows minimal decomposition with relatively high densification values.

The DC magnetic susceptibility curves that are normalized by the maximum superconducting values for ex situ - 800 °C - 15 min measured to ex situ - 1000 °C - 15 min samples are shown in Figure 4a, and the dwell time-varied ex situ-850 °C - 1 min to ex situ - 850 °C - 15 min samples are shown on the Figure 4b. A polycrystalline MgB_2_ sample superconducting transition curve is also included as reference. The onset of *T*_c_ for Polycrystalline MgB_2_ is determined to be 38.5 K with transition width (Δ*T*_c_) of 0.43 K. All samples show sharp superconducting transitions at the onset value of >38 K with Δ*T*_c_ of <1 K), indicating the high quality nature of the fabricated MgB_2_ bulks. Presently, our *T*_c_ results showed a negligible reduction in the onset value of bulk samples compared to the polycrystalline sample. The width of the superconducting transitions Δ*T*_c_ of all the samples was observed to slightly decrease in both the ex situ and in situ processes, as the processing temperature decreases at a controlled dwell time. The gradual degradation of the *T*_c_ could have resulted from the impurity phases. This was supported by Yakinci et al. [33].

The field dependence of *J*_c_(*B*) of the MgB_2_ samples at 20 K was estimated utilizing the extended Bean critical state model [29]. Figure 5a shows the variation of *J*_c_ curves of MgB_2_ bulk ex situ samples synthesized at different sintered temperatures of 800 °C, 850 °C, 900 °C, and 1000 °C. Self-field *J*_c_ [*J*_c_(0)] values for the samples are determined to be 401, 493, 413, and 333 kA/cm^2^, respectively. The *J*_c_(0) value for the Polycrystalline MgB_2_ sample is 298 kA/cm^2^, which clearly shows that the MgB_2_ bulk samples support carrying large *J*_c_ compared to the sample produced by the conventional sintering method. Among, the SPS-processed MgB_2_ samples, ex situ - 850 °C - 15 min superconductor was observed to be superior in the *J*_c_ performance, which could be attributable to the high density along with the relatively lower fraction of the secondary phase formation, as indicated on the sample XRD. Moreover, even though the densification values are higher in the case of the 900 °C and 1000 °C sintered samples, *J*_c_ performance is inferior and worsened as the sintering temperature increased. This is mainly due to the MgB_2_ phase degradation into fractions of non-superconducting phases. To further control this effect, we tried to control the regime of the sintering process. The *J*_c_(*H*) curves determined at 20 K for the ex situ - 850 °C - 1 min to ex situ - 850 °C - 15 min superconductors are shown in Figure 5b. The self-field *J*_c_ of the ex situ- 850 °C - 1 min, ex situ - 850 °C - 5 min, ex situ - 850 °C - 10 min, and ex situ - 850 °C - 15 min superconductors are determined to be 420 kA/cm^2^, 495 kA/cm^2^, 517 kA/cm^2^, and 493 kA/cm^2^, respectively. The MgB_2_ ex situ bulk synthesized at 850 °C for 10 min exhibited superior field dependence J_c_ performance. Presently, our results are superior to the reports available in the literature on the MgB_2_ bulks produced by SPS and HP [28,32]. The superior field dependence properties of the ex situ - 850 °C - 15 min superconductor could be attributable to the optimal sintering temperature and controlled dwell time, at which the phase degradation was minimal with a corresponding improvement in the density. With this, we successfully fabricated highly dense MgB_2_ bulk samples via the ex situ SPS method, and, mainly, such high densification values are achieved at lower sintering temperatures.

### 3.2. In Situ Synthesis

In the above sections, we produced MgB_2_ bulk superconductors with high mass density supporting large critical currents via the ex situ SPS method. However, in order to reduce the cost of the application and the time, we attempted to synthesize the MgB_2_ bulk samples via the in situ SPS method by mixing the stoichiometry ratios of Mg and B. This also aimed at fabrication of the bulk with abundant amounts of grain connectivity along with high mass density due to the in situ reaction, which could support even larger *J*_c,S_ in the MgB_2_ bulk samples. The in situ process for the phase analysis is shown in Figure 6, and the observed XRD was compared to our previously fabricated bulk sample using the in situ solid state sintering technique by the tubular furnace at 775 °C for 3 h.

At first, we tried to synthesize the in situ MgB_2_ bulk by SPS using the same temperature condition but with a longer dwell time of 20 min. With the occurrence of decomposition at this condition, we proceeded by slightly decreasing the dwell time and controlling the sintering temperatures, which was found to be effective for decreasing the deterioration of the MgB_2_ phase; however, it was necessary to find the optimum temperature for high-density conditions, at which we could also attain an equilibrium for the superconducting performance of our bulk in situ samples. The superconducting transitions in the MgB_2_ bulk synthesized by the in situ process are shown in Figure 7. The Figure 8 shows the superconducting field performances of the *J*_c_ of the in situ process.

Besides the contributions of secondary particles, one of the important observations we made was the significant increase in the SPS in situ sample density, by sole temperature control and grain size refinement; while keeping the initial pressure of 50 MPa, we could achieve ~90% density. This could be a breakthrough when further optimization by pressure increase is employed, since this could exceed the findings of Prikhna et al. [31] for the highest density obtained by the SPS in situ process and, hence, the superconducting performance and mechanical properties.

More crystallinity was observed for the in situ samples than the ex situ samples, which is evident in the degree of sharpness for the superconducting transitions. The observed suppression in the ex situ samples *T*_c_ could be due to the effect of the impurity phases, which occurred in the XRD analysis. This observation on the slight suppression in the *T*_c_ was supported by Dancer et al. [34]. This suggests that the lower processing temperatures could have attained a degree of sufficiency in recovering the disorder that may have existed in the initial (as received) powders.

All the samples tend to show improvement in the field performance, when compared to the bulk synthesized via tubular furnace; however, the in situ processed bulk sample at 750 °C was optimal at 504 kA/cm^2^ at 88% packing density for these sets of samples. This is so promising, when compared to the recent reports of Noudem et al. [28] for in situ processing at 500 kA/cm^2^ and 84% packing density. We suggest that the field dependence of the respective *J*_c_ varies with temperature optimization as a function of the grain size refinement and, hence, the densities. The additional pinning centers in this case may also have resulted from the nano-sized particles, which also maximize the flux pinning performance exhibited by our best sample bulks; meanwhile, we also proposed that the relevance of the *J*_c_ improvement is also attributed to the contribution of the refinement of the grains, which improved the flux pinning performance and consequently the *J*_c_.

### 3.3. Microstructural Analysis

The grain size and grain connectivity are crucial for determination of the *J*_c_ in polycrystalline superconductors. Microstructural characterization carried out via FESEM on the ex situ - 850 °C - 10 min and in situ - 750 °C - 15 min samples are compared with Polycrystalline MgB_2_ sample in Figure 9a–c, respectively. The statistical analysis of the grain size is completed using the ImageJ program, and the corresponding size histograms are shown in Figure 9d–f, respectively. The mean size of the MgB_2_ grains for polycrystalline MgB_2_, ex situ - 850 °C - 10 min, and in situ - 750 °C - 15 min samples, are determined to be 112 nm, 119 nm, and 132 nm, respectively. These results vindicate that the MgB_2_ bulk samples produced via SPS contain slightly finer sized MgB_2_ superconducting grains than the polycrystalline MgB_2_ sample. It is interesting to note here, even though both the ex situ - 850 °C - 10 min and in situ - 750 °C - 15 min samples are fabricated by the SPS method, the frequency of fine-sized grains is larger in the case of the former sample. This could be as a result of the further and careful optimization of the dwell time during the ex situ processing, which could have further controlled the grain sizes.

In order to complement our FESEM observations, we further conducted microstructural analyses via TEM investigations, as shown in Figure 9, on ex situ - 850 °C - 10 min and in situ - 750 °C - 15 min samples. We could suggest the occurrence of the nano-size MgO particles, which may have enhanced the flux pinning of the SPS processed samples, implying that the interactive contributions of the refined grains and the MgO inclusions could have been the primary source of the improved superconducting performance. In ex situ SPS processing, our observation for the optimized sample indicated the presence of nano-size grains of MgB_4_, as shown in Figure 10a, which could have been a contributor to supporting the superconducting *J*_c_. The TEM results for these samples in Figure 10a showed more refined grains of MgB_4_ as a secondary phase and, also, what could possibly be the inclusion of the MgO particles. We suppose that the dramatic increase in the superconducting *J*_c_ is attributed to this effect. As for the in situ process, the microstructure reveals the occurrence of the MgO system within the MgB_2_ matrix for the optimized conditions, but more grains tend to manifest with a shorter dwell time, resulting in better connectivity and more controlled inclusion of the MgO, as illustrated in Figure 10b.

Our findings indicate that both the SPS-processed bulk samples are superior compared to the polycrystalline sample. It could be emphasized here that the grain refinement, by virtue of the grain sizes, connectivity, and the resulted improved densities, aided in enhancing the *J*_c_. Among bulk samples, they exhibit different flux pinning behaviors as the slopes of the *J*_c_ curves are different at different fields. This clearly indicates that the clean grain boundary and lower porosity, along with the relatively higher densification, are supporting high *J*_c,S_ at moderate-applied magnetic fields, along with the nano-sized secondary phases, as observed from TEM. With this, we successfully synthesized the MgB_2_ bulk samples via the in situ method by SPS in its pure phase. Present self-field results are slightly inferior compared to ex situ bulks. However, it should be noted here that the in situ bulks are of low densification compared to ex situ, but still exhibit higher quality the bulk samples compared to a recent report [28]. By further optimization of the processing parameters, high-quality MgB_2_ bulks can be fabricated, which support even superior field dependence properties compared to any other method. In our case, the processing temperature and dwell time were the variable parameters considered to be effective.

### 3.4. Flux Pinning Analysis

To support our discussions regarding the TEM analysis, the pinning force, *f*_p_, was investigated using a scaling method based on the body of literature data and then compared to our studies. We calculated the scaling behavior and the best fits to the theory using the scaling methods of Dew-Hughes [35] and Kramer [36]. Flux pinning of the fabricated MgB_2_ bulk samples is scaled using the Dew-Hughes general expression.
*f*_p_ = *A* · (*h*)*^p^* · (1 − *h*) *^q^*(2)

Constants *p* and *q* are describing the actual pinning mechanism. The normalized flux pinning force density is denoted by *f*_p_ = *F*_p_/*F*_p,max_, and the reduced magnetic field is *h* = *H/H*_irr_. The value of the irreversibility field, *H*_irr_, was estimated as the field where *J*_c_ decreased to 100 A/cm^2^ at 20 K, which is customary in our works. The normalized pinning force *F*_p_/*F*_p,max_ versus the reduced fields were plotted, and the curves are shown in Figure 11. According to Dew-Hughes, the peak position of the *f*_p_(*h*) dependence indicates the type of pinning in the superconducting material. As per the model, the peak position of *f*_p_ vs. *h* at 0.2 implies grain boundary pinning, meanwhile 0.33 implies core, *δT*_c_ pinning, and so on. This flux pinning behavior was also supported by the works of Koblischka et al. [37] on the pinning force scaling analysis for high-*T_c_* superconductors. The resulting peak positions, *h*_o_, were found around ~0.25 and ~0.28, respectively, for the optimized in situ and ex situ samples, with a slight shift to the right towards higher field that supports the indication for particle pinning. The implication here is that some of contributions by the flux pinning may have resulted from nano-sized inclusions of the secondary phases; this assertion is also supported in the TEM analysis discussed in Figure 10. However, the polycrystalline bulk sample tends to exhibit the ideal peak position, which is around 0.22, indicating the effect of grain boundaries for vortex pinning. The shifts in the peak positions could also be attributed to the improved connectivity and the resulting refined grains, which added to our high suspicion of the temperature control and dwell time during synthesis. The mechanism for pinning other than grain boundaries for MgB_2_, was emphasized by other reports [38], which also supports our findings. The Figure 11 shows the flux pinning diagram showing peak positions for the optimized in situ and ex situ samples compared with polycrystalline MgB_2_.

## 4. Conclusions

We discussed the influence of the sintering temperature and dwell time on *J*_c_-*B* properties of dense in situ and ex situ MgB_2_ bulks synthesized by SPS. However, it is obvious that the *J*_c_-*B* characteristics improved as a result of appropriate nano-inclusions of MgO and MgB_4_ impurity phases in the final microstructure, which resulted in significant flux pinning enhancement. For in situ and ex situ, the best samples demonstrated self-field *J*_c_ behaviors at 504 and 517 kA/cm^2^, respectively. The good grain refinement and connection, which also influenced the high bulk densities, may have resulted in more improved *J*_c_ behaviors. Nevertheless, the ex situ synthesis is complicated and time-consuming, despite the challenges in controlling the impurity phases. We proposed that it might be appropriate to consider the in situ processing method of synthesizing high density MgB_2_ bulk, which is less complicated and saves time with its ease of control for the impurity phase. Meanwhile, more progress can be achieved for the SPS in situ processing by improving the quality of starting powders and applying high-pressure sintering. Our approach also supports the possibilities of other processing windows in which well-sintered MgB_2_ cores in wires and bulk magnets can be fabricated for practical applications.

## Figures and Tables

**Figure 1 materials-14-07395-f001:**
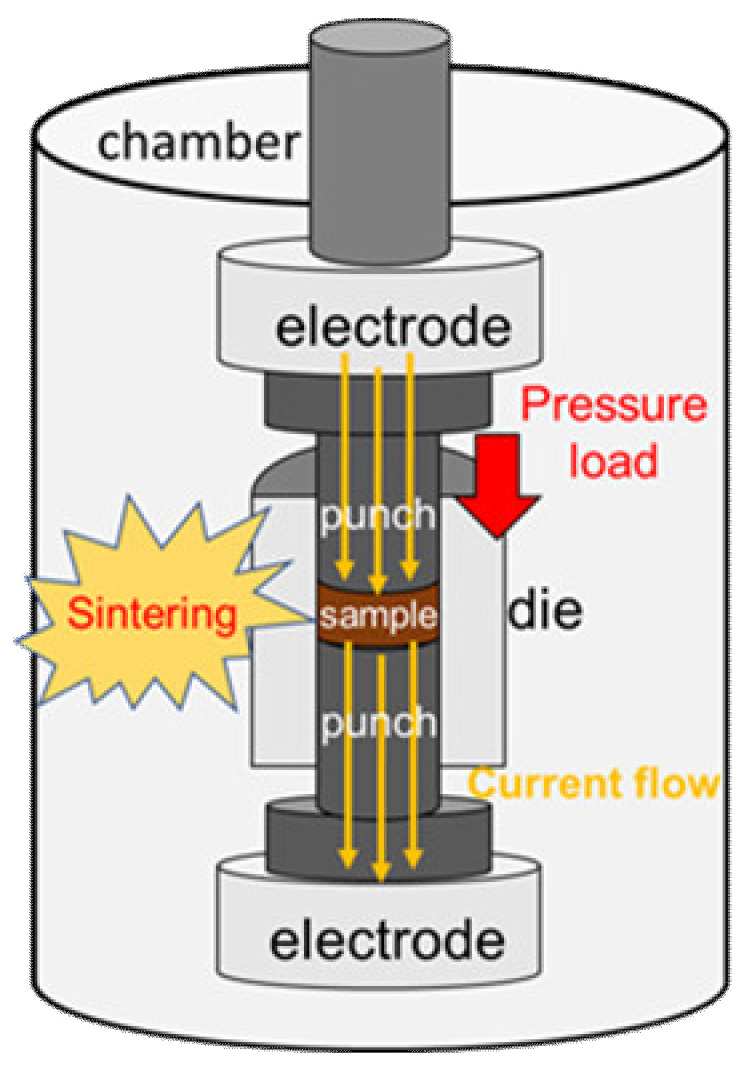
Schematic diagram of an SPS furnace.

**Figure 2 materials-14-07395-f002:**
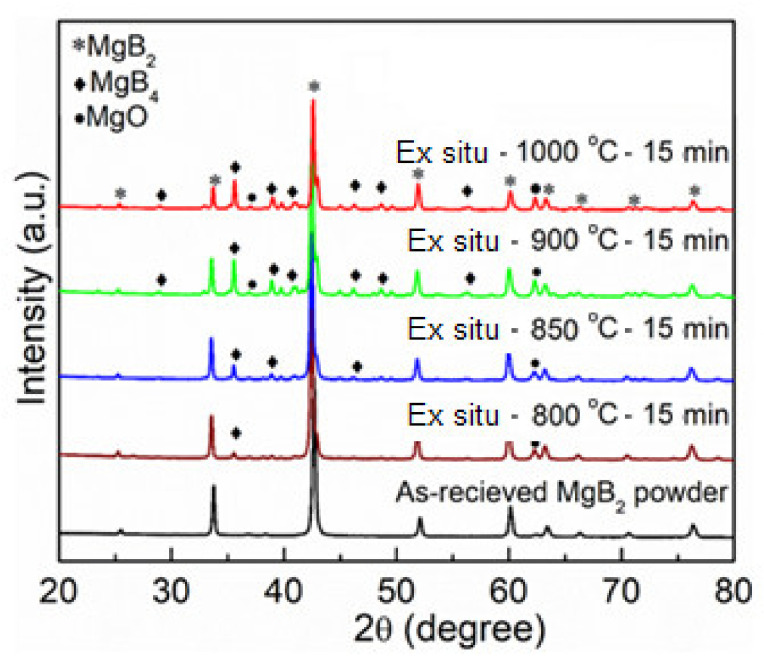
X-ray diffraction patterns for sintering temperature optimization of spark plasma sintered MgB_2_ bulk via the ex situ method.

**Figure 3 materials-14-07395-f003:**
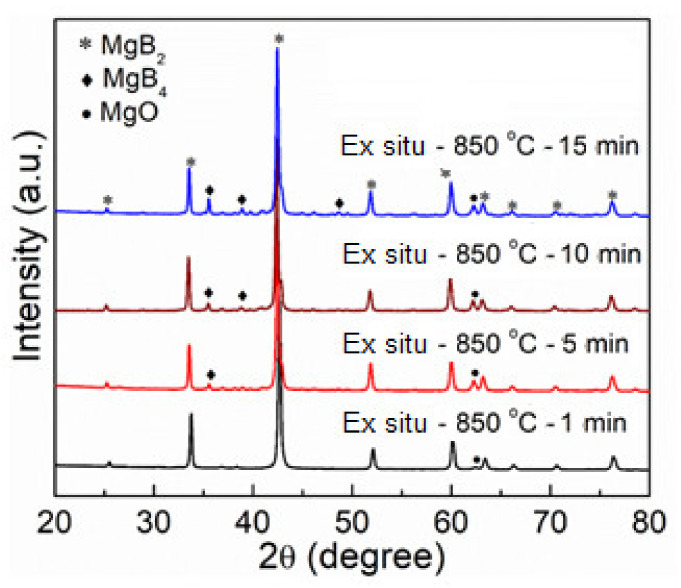
The X-ray diffraction patterns for MgB_2_ ex situ further optimization at 850 °C.

**Figure 4 materials-14-07395-f004:**
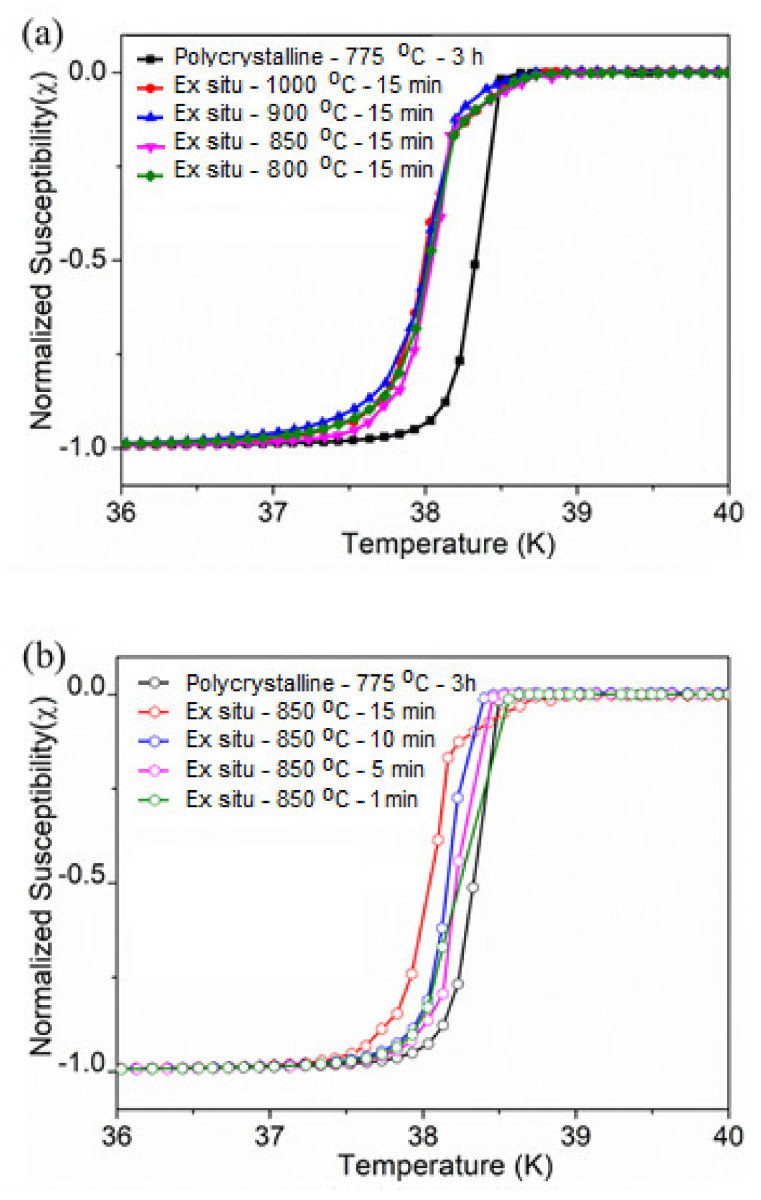
Superconducting transition in the bulk MgB_2_ processed by SPS via ex situ (**a**) variation of the sintering temperature and (**b**) variation of the dwell time.

**Figure 5 materials-14-07395-f005:**
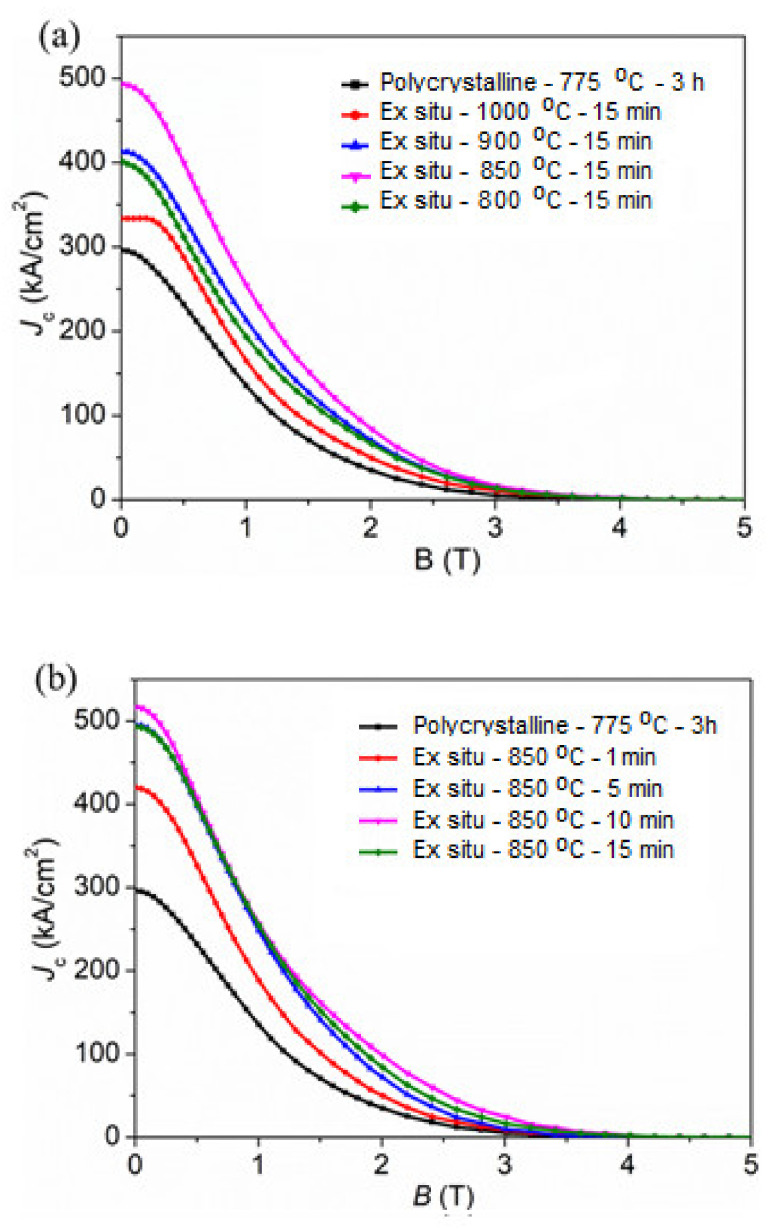
The magnetic field dependence of *J*_c_ curves determined at 20 K for MgB_2_ bulk superconductors fabricated by the SPS ex situ method for (**a**) sintering temperature variation and (**b**) dwell time variation.

**Figure 6 materials-14-07395-f006:**
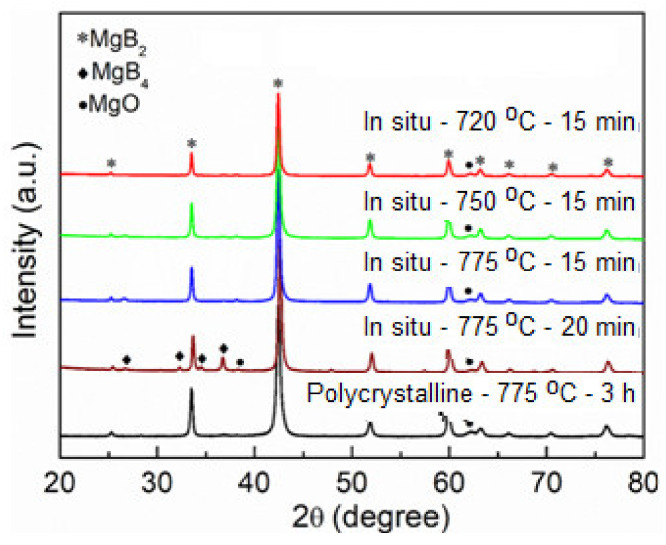
X-ray diffraction patterns for spark plasma sintered MgB_2_ bulk via the in situ method.

**Figure 7 materials-14-07395-f007:**
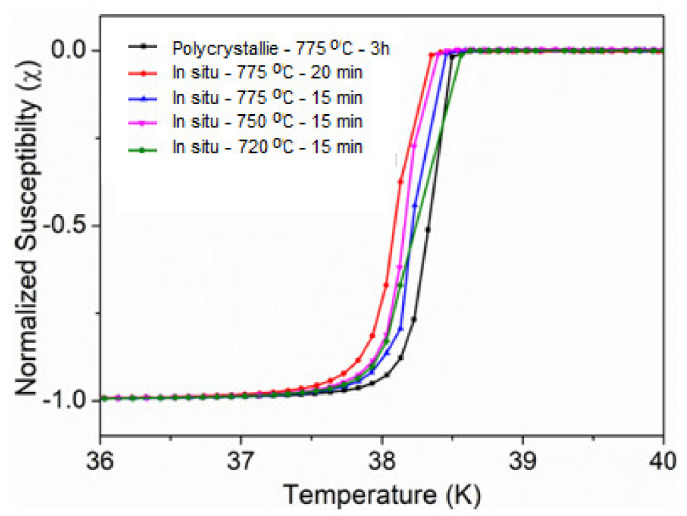
The superconducting transitions in the MgB_2_ bulk synthesize by in situ process.

**Figure 8 materials-14-07395-f008:**
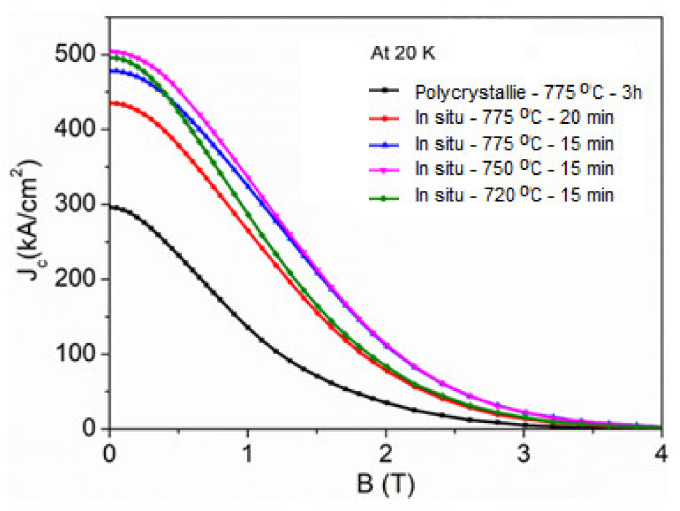
Shows the superconducting field performances of the *J*_c_ of the in situ process.

**Figure 9 materials-14-07395-f009:**
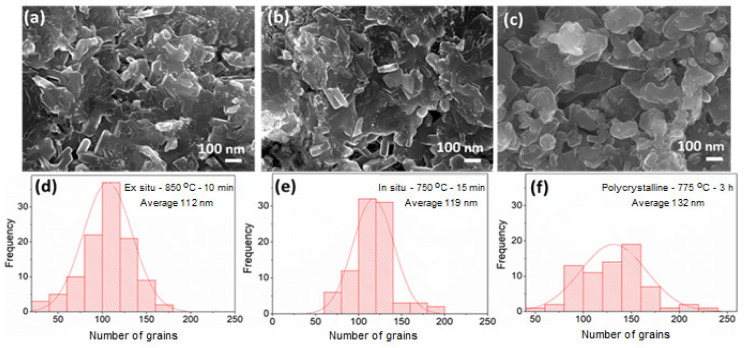
High magnification (30.000×) FE-SEM images for optimized fractured bulk samples produced by (**a**) SPS ex situ, (**b**) SPS in situ, and (**c**) polycrystalline MgB_2_. The subfigures (**d**–**f**) shows the statistical analysis of the grain size distributions.

**Figure 10 materials-14-07395-f010:**
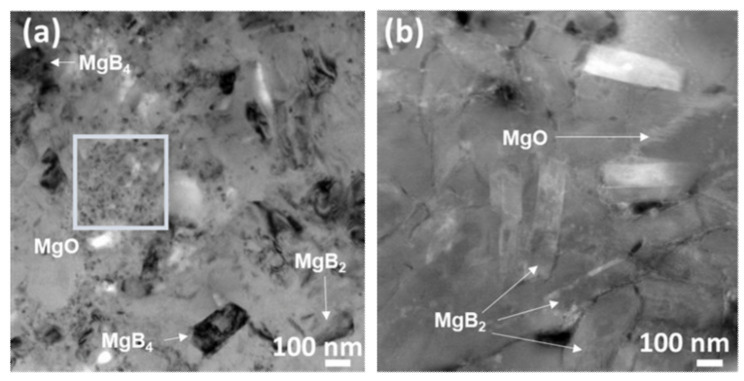
TEM micrographs of optimized polished surfaces of (**a**) SPS ex situ and (**b**) SPS in situ.

**Figure 11 materials-14-07395-f011:**
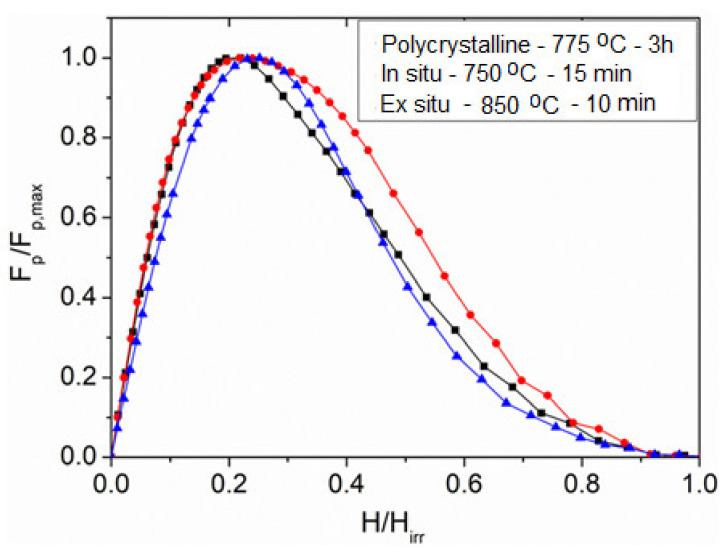
The flux pinning diagram showing peak positions for the optimized in situ and ex situ samples compared with polycrystalline MgB_2_. There is a slight shift in peak positions to the right.

**Table 1 materials-14-07395-t001:** The polycrystalline MgB_2_ sample density was determined to be 1.73 g/cm^3^, and the corresponding relative densification value is 66%. The density of the ex situ - 800 °C - 15 min–ex situ-1000 °C - 15 min samples was found to be 2.41–2.59 g/cm^3^, which corresponds to 92–99% of the theoretical density of the same compound. The densification values for dwell time, relative density, and superconducting transition (*T*_c(onset)_, *T*_(offset)_, Δ*T*_c_) for samples prepared by the ex situ and in situ methods. The standard deviation(s) for *T*_c,S_ and Δ*T*_c,S_: σ < 0.21.

Sample	Dwell Time (min)	Relative Density (%)	Superconducting Transition (K) *T*_c (onset)_ *T_c (_*_offset)_ Δ*T*_c_			Phase
	Ex Situ					
Ex situ_1000 °C	15	99	38.3	37.7	0.6	Decomposition present
Ex situ_900 °C	15	99	38.2	37.7	0.5	Decomposition present
Ex situ_850 °C	15	97	38.2	37.7	0.4	Decomposition reduced
Ex situ_800 °C	15	92	38.1	37.7	0.4	Little decomposition
Ex situ_850 °C	10	97	38.3	38.0	0.3	Improved phase/little MgO
Ex situ_850 °C	5	89	38.1	37.8	0.3	Improved phase/little MgO
Ex situ_850 °C	1	86	38.5	37.9	0.6	Little MgO
	In Situ					
In situ_775 °C	Tubular/3 h	66	38.5	38.1	0.3	Little MgO
In situ_775 °C	SPS/20	90	38.3	38.0	0.3	MgO/MgB_4_
In situ_775 °C	SPS/15	90	38.4	38.2	0.2	Little MgO
In situ_750 °C	SPS/15	88	38.3	38.1	0.3	Little MgO
In situ_720 °C	SPS/15	83	38.5	38.3	0.2	Little MgO

## Data Availability

The data presented in this study are available on request from the corresponding author.

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
