# Peer review of "Synthesis of Dense MgB2 Superconductor via In Situ and Ex Situ Spark Plasma Sintering Method"

_materials, 2021, doi:10.3390/ma14237395_

Round 1

Reviewer 1 Report

     The Authors of the current manuscript investigated the relationship between sintering conditions and Jc-B using ex-situ (sintering MgB2 synthesized powder) and in-situ (reaction sintering of Mg and B powder) methods. According to them higher density with suppressed particle growth and suppression of the formation of coarse particles of MgB4 and MgO were found to be effective in improving the Jc-B characteristics. Microstructural characterization and flux pinning analysis revealed the possibility of refined MgO inclusions and MgB4 phase as new pinning centers which greatly contributed to the Jc-B properties. I trust in the results presented in the current manuscript. I recommend publication of the current manuscript however, I suggest to introduce minor changes:

- Figure 4; Description of x-axis in both panels is “Temperature  (K) 2theta (degree)”; most likely it should be “Temperature (K)”;

- Figures 5 and 8: x- axis is described in SI units while y-axis is described in CGS units; it is better to not mix units;

- Figures 2, 3, and 6: y-axis is described in “(a. u)” units; should be “(a. u.)”;

- Line 98 : is Hc; should be Hc2.

Author Response

Dear Reviewer,

Thank you very much for your comments and your time. We considered your suggested comments and made improvements to the figures in the manuscript. We sincerely hope you are satisfied with the detailed responses to the comments.

The Authors of the current manuscript investigated the relationship between sintering conditions and Jc-B using ex-situ (sintering MgB2 synthesized powder) and in-situ (reaction sintering of Mg and B powder) methods. According to them higher density with suppressed particle growth and suppression of the formation of coarse particles of MgB4 and MgO were found to be effective in improving the Jc-B characteristics. Microstructural characterization and flux pinning analysis revealed the possibility of refined MgO inclusions and MgB4 phase as new pinning centers which greatly contributed to the Jc-B properties. I trust in the results presented in the current manuscript. I recommend publication of the current manuscript however; I suggest to introduce minor changes:

- Figure 4; Description of x-axis in both panels is “Temperature (K) 2theta (degree)”; most likely it should be “Temperature (K)”;

            Thank you very much, we have included the corrected figures in the manuscript.

- Figures 5 and 8: x- axis is described in SI units while y-axis is described in CGS units; it is better to not mix units;

            These Figures show SI units and are commonly used in the literature, e.g. a Figure showing the dependence of the critical current as a function of the magnetic field can be found in the paper: Supercond. Sci. Technol. 27 (2014) 044007, pp. 4.

- Figures 2, 3, and 6: y-axis is described in “(a. u)” units; should be “(a. u.)”;

            Thank you very much, the amendment are marked in the text of the manuscript

- Line 98: is Hc; should be Hc2.

            Thank you very much, the amendment are marked in the text of the manuscript

Reviewer 2 Report

Strengths

Authors discussed the influence of sintering temperature and dwell-time on Jc-B properties of dense in-situ and ex-situ MgB2 bulks synthesized by SPS. However, it is obvious that the  Jc-B characteristics improved as a  result of appropriate nano inclusions of MgO and MgB4 impurity phases in the final microstructure, which resulted in significant flux pinning enhancement. In-situ and ex-situ, the best samples demonstrated self-field Jc behaviors at 504 and 516 kA/cm2, respectively. Meanwhile, more progress can be achieved for SPS in-situ processing by improving the quality of starting powders and applying high-pressure sintering. Our approach also supports the possibilities of other processing windows in which well-sintered MgB2 cores in wires and bulk magnets can be fabricated for practical applications.

Weakness

Unfortunately, there are unfortunate errors and typographical errors.

  1. In table 1, you need to change the name.
  2. It is necessary to remove 2θ (degree) on the abscissa axis in Figure 4 (a,b).
  3. The link to Figure 6 should be earlier than the figure.
  4. Increase the size of the letters in lines 300-302 and 417, 418.
  5. The unfortunate name of the ordinate axes in Figure 9. It is better to name the Number of grains.
  6. …determined to be 104 nm, 118 nm and 137 nm respectively (line 341). Figure 9 (d,e,f) shows 112 nm, 119nm, 132 nm respectively?
  7. In-situ and ex-situ, the best samples demonstrated self-field Jc behaviors at 504 and 516 kA/cm2, respectively. (Line 427). It will be right: ….. Jc behaviors at 504 and 517 kA/cm2, respectively.
  8. Update references.

Author Response

Thank you very much for the Reviewer's comments. We considered all the remarks indicated in the reviewed manuscript. The authors assume that the responses to remarks will be satisfactory. Below are the detailed answers.

Authors discussed the influence of sintering temperature and dwell-time on Jc-B properties of dense in-situ and ex-situ MgB2 bulks synthesized by SPS. However, it is obvious that the Jc-B characteristics improved as a result of appropriate nano inclusions of MgO and MgB4 impurity phases in the final microstructure, which resulted in significant flux pinning enhancement. In-situ and ex-situ, the best samples demonstrated self-field Jc behaviors at 504 and 516 kA/cm2, respectively. Meanwhile, more progress can be achieved for SPS in-situ processing by improving the quality of starting powders and applying high-pressure sintering. Our approach also supports the possibilities of other processing windows in which well-sintered MgB2 cores in wires and bulk magnets can be fabricated for practical applications.

 MgB2 plays an important role in practical applications due to its low cost, ease of manufacture, and relatively high critical current. We agree that in-situ SPS treatment can be improved by improving the quality of the starting powders and applying high-pressure sintering. Our method presented in the manuscript draws attention to the quite unique problem of fine-tuning the impurity phase during synthesis.

Unfortunately, there are unfortunate errors and typographical errors.

  1. In table 1, you need to change the name.

                Thank you very much, We have included the amendment in the text of the manuscript.

            "The polycrystalline MgB2 sample density was determined to be 1.73 g/cm3          corresponding relative densification value is 66%. The density of the ex-situ - 800 °C -     15 min ÷ ex-situ - 1000 °C - 15 min samples was found to be 2.41 g/cm3 ÷ 2.59 g/cm3,    which corresponds to 92 ÷ 99 % of theoretical density of the same compound. The        densification values for dwell time, relative density and superconducting transition         (Tc(onset), T(offset), ΔTc) for samples prepared by ex-situ and in-situ methods are given in             Table."

  1. It is necessary to remove 2θ (degree) on the abscissa axis in Figure 4 (a,b).

Thank you very much, we have included the corrected figures in the manuscript.

  1. The link to Figure 6 should be earlier than the figure.

            Thank you very much, the amendment included in the manuscript.

  1. Increase the size of the letters in lines 300-302 and 417, 418.

                Thanks for paying attention, we have improved it.

  1. The unfortunate name of the ordinate axes in Figure 9. It is better to name the Number of grains.

Thank you, we have corrected the caption on the horizontal axis of the drawing. Now there is "Number of grains".

  1. …determined to be 104 nm, 118 nm and 137 nm respectively (line 341). Figure 9 (d,e,f) shows 112 nm, 119nm, 132 nm respectively?

            Thank you, we have corrected it.

  1. In-situ and ex-situ, the best samples demonstrated self-field Jc behaviors at 504 and 516 kA/cm2, respectively. (Line 427). It will be right: ….. Jc behaviors at 504 and 517 kA/cm2, respectively.

            Thank you, we have corrected it.

  1. Update references

     Two references have been added to the manuscript in the places marked in the text:

     [35] Dew-Hughes, D., Flux pinning mechanisms in type-II superconductors, Philos.          Mag. 1974, 30, 293.

     [36] Kramer, E.J., Scaling laws for flux pinning in hard superconductors, J. Appl. Phys. 1973, 44, 1360.

     Reference title was completed:

     [37] Koblischka M.; and Muralidhar, M., Pinning force scaling analysis of Fe-based high-Tc superconductors, Inter. J. Mod. Phys. B 2016, 30, 1630017.

Reviewer 3 Report

The authors present how the synthesis method by applying spark plasma sintering (SPS) improve the critical current of MgB2 and the related impurity pinning effect on the properties of the compound. Although MgB2 is a standard BSC superconductor, it plays an important role in practical applications due to its low cost, easy to manufacture, and relatively high critical current. However, it’s very difficult to improve Tc of MgB2 by many techniques. On the other hand, it’s valuable to increase the critical current for the application purpose. The authors’ method is a good way to tuning the impurity phase during the synthesis. The experiment and procedures are solid and scientific sound. I would potentially accept to publish it in the Materials after the authors answer my questions properly and make some minor revision on the manuscript.

  1. The abstract is not clear. Why do you mention the “uniaxial pressure” in the very beginning but there are a few words in the context to mention it? If it’s not a vital procedure, it should not be present here.
  2. Furthermore, the authors actually use the wrong terminology of “uniaxial pressure” in Line 145. Uniaxial pressure means applying stress in one direction of a sample while the perpendicular direction of the sample is free. But in the context, the authors just use a standard pressure technique which is definitely not uniaxial.

Typos:

  1. Some characters, e.g. Line 130, 147, 182, etc. I guess the authors mean the “-“ rather than the “ ÷ ”
  2. Line 137, useless “(II)”

Author Response

Thank you very much for the Reviewer's comments. We considered all the remarks indicated in the reviewed manuscript. The authors assume that the responses to remarks will be satisfactory. Below are the detailed answers.

The authors present how the synthesis method by applying spark plasma sintering (SPS) improves the critical current of MgB2 and the related impurity pinning effect on the properties of the compound. Although MgB2 is a standard BSC superconductor, it plays an important role in practical applications due to its low cost, easy to manufacture, and relatively high critical current. However, it’s very difficult to improve Tc of MgB2 by many techniques. On the other hand, it’s valuable to increase the critical current for the application purpose. The authors’ method is a good way to tuning the impurity phase during the synthesis. The experiment and procedures are solid and scientific sound. I would potentially accept to publish it in the Materials after the authors answer my questions properly and make some minor revisions on the manuscript.

  1. The abstract is not clear. Why do you mention the “uniaxial pressure” in the very beginning but there are a few words in the context to mention it? If it’s not a vital procedure, it should not be present here.
  2. Furthermore, the authors actually use the wrong terminology of “uniaxial pressure” in Line 145. Uniaxial pressure means applying stress in one direction of a sample while the perpendicular direction of the sample is free. But in the context, the authors just use a standard pressure technique which is definitely not uniaxial.

                Thanks for your attention, we have removed the word "uniaxial".

Typos:

  1. Some characters, e.g. Line 130, 147, 182, etc. I guess the authors mean the “- “rather than the “÷”

Thanks for your suggestion, we have corrected these markings in the manuscript.

  1. Line 137, useless “(II)”

We removed the marking "(II)", and bolded the individual stages.
